# Therapeutic Drug Monitoring of Vancomycin Concentrations for the Management of Bone and Joint Infections: An Urgent Need

**DOI:** 10.3390/tropicalmed8020113

**Published:** 2023-02-13

**Authors:** Laura Rio-No, Luisa Sorli, Alba Arderiu-Formenti, Marta De Antonio, Lucas Martorell, Isaac Subirana, Lluis Puig, Albert Alier, Joan Gómez-Junyent, Daniel Pérez-Prieto, Sonia Luque

**Affiliations:** 1Pharmacy Department, Hospital del Mar, Parc de Salut Mar, 08003 Barcelona, Spain; 2Infectious Diseases Department, Hospital del Mar, Parc de Salut Mar, 08003 Barcelona, Spain; 3Infectious Pathology and Antimicrobials Research Group (IPAR), Institut Hospital del Mar d’Investigacions Mèdiques (IMIM), 08003 Barcelona, Spain; 4Spanish Network for Research in Infectious Diseases (REIPI RD 16/0016/0015), Instituto de Salud Carlos III, 28029 Madrid, Spain; 5Study Group on Osteoarticular Infections of the Spanish Society of Clinical Microbiology and Infectious Diseases (GEIO-SEIMC), 28003 Madrid, Spain; 6Faculty of Health and Life Sciences, Universitat Pompeu Fabra (UPF), 08003 Barcelona, Spain; 7Orthopedic Surgery and Traumatology Department, Hospital del Mar, Parc de Salut Mar, 08003 Barcelona, Spain; 8Faculty of Medicine, Universitat Autònoma de Barcelona (UAB), 08193 Barcelona, Spain; 9CIBER en Enfermedades Cardiovasculares (CIBERCV), 28029 Madrid, Spain; 10Institut Hospital del Mar d’Investigacions Mèdiques (IMIM), 08003 Barcelona, Spain

**Keywords:** bone and joint infections, vancomycin, therapeutic drug monitoring, pharmacokinetics, nephrotoxicity, dosage regimens

## Abstract

Vancomycin is used for the treatment of bone and joint infections (BJI), but scarce information is available about its pharmacokinetic/pharmacodynamic (PK/PD) characteristics. We aimed to identify the risk factors associated with the non-achievement of an optimal PK/PD target in the first therapeutic drug monitoring (TDM). Methods: A retrospective study was conducted in a tertiary hospital from January 2020 to January 2022. Patients with BJI and TDM of vancomycin on day 2 of treatment were included. Initial vancomycin fixed doses (1 g every 8 h or 12 h) was decided by the responsible doctors. According to TDM results, dosage adjustments were performed. An AUC_24h_/MIC < 400 mg × h/L, between 400 and 600 mg × h/L and >600 mg × h/L, were defined as suboptimal, optimal and supratherapeutic, respectively. Patients were grouped into these three categories. Demographic, clinical and PK characteristics were compared between groups. Nephrotoxicity at the end of treatment was assessed. Results: A total of 94 patients were included: 22 (23.4%), 42 (44.7%) and 30 (31.9%) presented an infratherapeutic, optimal and supratherapeutic PK/PD targets, respectively. A younger age and initial vancomycin dose <40 mg/kg/day were predictive factors for achieving a suboptimal PK/PD target, while older age, higher serum-creatinine and dose >40 mg/kg/day were associated with overexposure. The nephrotoxicity rate was 22.7%. More than 50% of patients did not achieve an optimal PK/PD. Considering age, baseline serum-creatinine and body weight, TDM is required to readily achieve an optimal and safe exposure.

## 1. Introduction

Bone and joint infections (BJI) include a wide range of complex infections affecting bone and joints and are associated with a high morbidity and disability [1,2]. In addition, the aging of the population has increased the prevalence of these infections, causing an important increase in health costs.

BJIs are complicated infections due to the presence of bacterial biofilms [3] and the difficulty in achieving optimal antibiotic concentrations in bone tissues. In this setting, an aggressive combination treatment involving surgery and an optimum antibiotic therapy containing antibiotics with good bone penetration and antibiofilm activity is required. 

Despite the advances in recent years in the field of diagnosis and treatment, therapeutic failure is frequent in BJI, with rates up to 50% in some publications [4]. Gram positive cocci (GPC), mainly *Staphylococcus aureus* and coagulase negative Staphylococci, remain the most frequent isolated microorganisms. In this scenario, the current guidelines recommend the use of vancomycin as an option for empirical and targeted therapy [5,6], especially in countries with high prevalence of methicillin-resistant *S. aureus* (MRSA) and methicillin-resistant *Staphylococcus epidermidis* (MRSE). However, despite having been used for 60 years, vancomycin side effects continue to be a matter of concern and therapeutic drug monitoring (TDM) has been proposed as a useful strategy to maximize effectiveness and reduce renal toxicity [7]. According to the latest TDM guidelines, the PK/PD target of vancomycin is an area under the curve over 24 h to a minimum inhibitory concentration ratio (AUC_24h_/MIC) of 400–600 mg × h/L [8,9,10]. However, it should be highlighted that almost all data available on vancomycin PK/PD and efficacy and toxicity have been derived from patients who have been treated for serious infections such as from MRSA and/or complicated bloodstream infections, while data on other type of infections, such as BJI, are not available.

As BJI are difficult-to-treat infections because antibiotics need to penetrate through the rigid bone structure and into the synovial, the TDM of vancomycin must ensure an optimal exposure in the early phases of treatment [11,12].

To date, no study has previously evaluated the PK/PD achievement of vancomycin in treating BJI. The aim of this study was to identify the risk factors associated with the non-achievement of an optimal pharmacokinetic/pharmacodynamic (PK/PD) target of vancomycin in the first therapeutic drug monitoring (TDM) sample.

## 2. Materials and Methods

### 2.1. Study Design and Population

This was a retrospective cohort study conducted among hospitalized patients at Hospital del Mar (Barcelona, Spain), a 450-bed university hospital, from January 2020 to January 2022.

This study was approved by the Ethics Committee Parc de Salut Mar (approval number: 2022/10523), and the need for written informed consent was waived due to the retrospective nature of the study.

All adult patients who were treated with vancomycin for a suspected or confirmed BJI and were undergoing therapeutic drug monitoring (TDM) of the plasma concentrations on the second day of treatment were included. This is routinely performed in the TDM program of our center. The diagnosis of prosthetic joint infection, fracture-related infection and/or osteomyelitis were performed using the EBJIS criteria [13]. For the purpose of our analysis, these entities were further categorized as prosthetic joint infection and osteoarthritis with or without implant [14].

According to our hospital protocol, initial vancomycin doses ranged between 1 g every 8 h or 12 h combined with ceftazidime, 2 g every 8 h, and the first vancomycin dosage was decided by the responsible medical team. According to the TDM results, dosage adjustments were performed. Vancomycin was administered by 1 h intermittent infusion and blood samples were obtained on day 2 of treatment at pre-dose (trough concentration or C_min_) and 1 h after the end of the infusion (peak concentration or C_max_). After collection, samples were immediately transported to the laboratory, centrifuged and analyzed using the method of fluorescence polarization immunoassay (FPIA (TDX, Abbott Laboratories, North Chicago, IL, USA)).

All the individual PK parameters and the AUC_24h_ were calculated using the Bayesian forecasting model that contains a bicompartimental population model of vancomycin (Abbottbase pharmacokinetic system PKS^®^, Abbott Laboratories, PKS, Chicago, IL, USA). The program makes an individual estimation of the pharmacokinetics of the patient, based on the minimization of the sum of squared differences between the observed and predicted values. According to these parameters, predictions about the best dose to achieve AUC_24h_ and trough concentrations within the range can be made.

The PK/PD target of vancomycin according to the recently published guidelines was an AUC_24h_/MIC ≥400 and <600 mg × h/L as the efficacy and safety target, respectively [10], which corresponds to an AUC_24h_ of 400–600 mg × h/L for empirical therapy, assuming a MIC of 1 mg/L. Consequently, an AUC_24h_ <400 mg × h/L was considered suboptimal and >600 mg × h/L supratherapeutic throughout this study, as a MIC of 1 mg/L was used in all patients. However, the real MIC of the isolated pathogen, when available, was taken into consideration to make the dosage recommendations [10,15]. MICs were determined by a broth microdilution method.

Patients were grouped into 3 different categories: suboptimal, optimal and supratherapeutic, and demographic, clinical and PK characteristics were compared between groups. The predictive factors associated with the non-achievement of an optimal PK/PD target were evaluated for both cases (suboptimal and supratherapeutic PK/PD target). Nephrotoxicity at the end of treatment was assessed by the KDIGO criteria, and in the majority of patients, renal function parameters were monitored twice a week [16]. The Glomerular Filtration Rate (GFR) was estimated with the CKD-EPI equation.

### 2.2. Dosage Recommendations Based on TDM Results

According to the TDM results, dosage recommendations were performed in all patients after considering the AUC_24h_ value, the C_min_ value, patients’ renal function and clinical evolution.

During the first days of treatment, an AUC_24h_/MIC closer to the upper limit of the recommended value (600 mg × h/L) was sought in order to maximize plasma exposure due to the poor penetration of vancomycin into bone [17,18]. In addition, a C_min_ between 10 and 15 mg/L was avoided to prevent the emergence of resistance and reduce the risk of nephrotoxicity [7,10,19,20]. All dosage adjustments were made using the Abbottbase-Pharmacokinetic System (PKS^®^), a two-compartment Bayesian forecasting program.

### 2.3. Clinical Outcomes

Clinical failure was defined as the need for a second debridement in the 30 days after the first surgery, the change to another anti-GPC antibiotic (e.g., linezolid, daptomycin or ceftaroline) or the presence of vancomycin-associated toxicity during treatment. Other clinical outcomes such as the global clinical cure at the end of antibiotic treatment, the need of a suppressive antibiotic treatment and 30-day all-cause mortality were also registered.

### 2.4. Statistical Analysis

Quantitative data are expressed as median (range or interquartile range (IQR)), and categorical data as absolute frequencies (percentages). The comparison of continuous variables was performed using the one-way ANOVA test for those with a normal distribution and the Kruskal–Wallis test when normality could not be assumed. For dichotomous variables, the chi-square and the Fisher exact test were applied.

Logistic regression was used to explore factors associated with the non-achievement of a defined PK/PD target (AUC_24h_/MIC between 400 and 600 mg × h/L). Univariate analyses were performed separately for each of the risk factor variables to ascertain the odds ratio and 95% confidence interval (CI). Variables with a *p* value < 0.15 in the univariate analyses were included in the logistic regression model for the multivariate analysis. Finally, a backward stepwise selection process was used to select the final variables forcing age, sex and vancomycin dose level into the model.

Non-significant variables were assessed as confounding variables and had to have a meaningful change in the regression coefficients of other variables to remain in the model. The results of logistic regression analyses were reported as adjusted ORs with 95% CIs. The two obtained multivariate logistic regression models were assessed by the Hosmer and Lemeshow Goodness of Fit Statistic and the C-statistic representing the area under the ROC curve.

A 2-sided *p* value < 0.05 was considered statistically significant. The SPSS (SPSS, Chicago, IL, USA) version 24.0 statistical package was used throughout.

## 3. Results

A total of 94 patients with BJI were included, with 47 (50%) females and a median age of 68.5 (56.2–79.8) years. A total of 42 (44.7%) patients achieved the defined PK/PD target; in 22 (23.4%), the target was suboptimal, and in 30 (31.9%), it was considered supratherapeutic.

Demographic and clinical data of the three compared groups are shown in Table 1, whereas Table 2 includes the PK parameters of vancomycin. Figure 1 shows the distribution of the vancomycin daily dose values between the three groups of patients with a suboptimal, optimal and supratherapeutic PKPD target.

According to the microbiological results, vancomycin was used empirically in 49 patients, and in 45 (47.9%), Gram-positive bacteria were isolated, the most common being *S. epidermidis* in 14 (31.1%) patients, methicillin-sensible *S. aureus* (MSSA) in 13 (28.9%), MRSA in 4 (8.9%) patients and enterococci in 4 (8.9%). In 10 (10.6%) patients, a gram-negative microorganism was cultured. The MIC was available in 45 (47.9%) patients and the median value was 1 mg/L being >1 mg/L in 14 (31.1%) patients.

In the multivariate analysis using a logistic regression model, risk factors for achieving a suboptimal PK/PD target (AUC_24h_/MIC < 400 mg × h/L) were a younger age and a vancomycin initial loading dose lower than 35 mg/kg/day (Table 2).

On the other hand, the identified risk factors for having a supratherapeutic PK/PD target were the male gender, an older age, a lower body weight and a high baseline serum creatinine (Table 3), identified using a logistic regression model.

### 3.1. Clinical Evolution

Thirty-one (33%) patients were diagnosed with prosthetic joint infection, sixteen (17%) with osteoarthritis without implant, fifteen (16%) with osteoarthritis with implant and twenty-one (34%) with suspected prosthetic joint infection. Forty-six (74.2%) patients attained clinical cure, whereas seven (11.3%) patients were reported to have clinical failure. Eight (12.9%) patients lacked clinical follow-up, and one died (not related to the infection).

### 3.2. Nephrotoxicity at the End of Treatment

Nephrotoxicity at the end of treatment was assessed in 66 (70.2%) patients (no parameters of renal function at end of treatment were available in the other 28) and acute kidney injury was observed in 15 (22.7%) patients, but in all cases, it was stage 1 (an increase ≥0.3 mg/dL within 48 h or ≥1.5- to 2-fold from baseline).

### 3.3. TDM-Based Vancomycin Dosage Recommendations

Based on the TDM results, dose recommendations were performed in all patients, these being dose maintenance, dose increase and dose reduction in 32 (34%), 31 (33%) and 31 (33%) patients, respectively. All of them were accepted by the responsible clinician.

In 16 of the 42 patients (38.1%) that achieved an optimal PK/PD target, a dose modification was recommended: in 5, a dose reduction was recommended because the C_min_ was higher than 15 mg/L, and in 11, a dose increase was recommended because the AUC_24h_ value was close to the lowest limit of 400 mg × h/L. A median (range) number of two (1–8) dose adjustments based on TDM results were performed. In 68 patients (72.3%), a second TDM sampling could be performed, 50 of them (73.5%) being within the therapeutic range.

## 4. Discussion

To the best of our knowledge, this is the first study evaluating the performance of early TDM of vancomycin in patients with BJI. We observed that less than 50% of patients achieved an optimal exposure of vancomycin on day 2 of treatment after the administration of an initial vancomycin dose that amounted to 1 g every 8 h or 12 h. In about 20% of the patients, the PK/PD target (AUC_24h_/MIC) was considered suboptimal, and in the other third, supratherapeutic.

All these results suggest that it may be necessary to do TDM of vancomycin as soon as possible to be able to identify those patients needing an early dosage adjustment for these difficult-to treat infections.

However, it must be considered that the optimal PK/PD target of vancomycin for the treatment of BJI has not been elucidated and all evidence comes from patients with bacteriemia or pneumonia caused by MRSA [7]. In the previous guidelines, a trough value of 15–20 mg/L at a steady rate was suggested for difficult-to-treat infections [20], but this value has also been associated with a high risk of nephrotoxicity, and the debate is still open.

When we evaluated the predictive factors for achieving a suboptimal and a supratherapeutic PK/PD target, we observed that younger patients who received an initial vancomycin total daily dose <35 mg/kg/day were those with a high risk of having a suboptimal plasma exposure.

Based on these results, we suggest that this patient profile may need a higher initial vancomycin dose to rapidly achieve an optimal PK/PD target from the beginning of the treatment.

On the contrary, a worse baseline renal function with a higher age, male gender and a lower body weight were identified as predictors for having a supratherapeutic plasma exposure, while being female was a protective factor. Surprisingly, a high initial vancomycin dose could not be included in the model.

Consequently, male and old patients with a reduced renal function at the beginning of the treatment are those with a high risk of overexposure, especially those with low body weight (e.g., malnutrition), even at normal vancomycin doses.

Previous studies have identified renal function, age and body weight as factors influencing predictors of vancomycin plasma concentrations within non-critically ill patients [21,22,23,24], but only one of them performed TDM in an early phase of treatment [25], as we did. In addition, only our study was focused on patients with BJI and used the AUC_24h_/MIC as a PK/PD target instead of the C_min_ or trough concentration, as it has been recommended in the latest guidelines.

It is true that how vancomycin penetrates in bone and joints is not well known. Previous studies have shown that vancomycin reaches concentrations in bone that exceed the MIC90 against *S. aureus* (1 mg/mL) and *Streptococci* (2 mg/mL) and the MIC50 of enterococci (1 mg/mL), but perhaps not the MIC90 of enterococci (MIC > 16 mg/mL) [18,26,27]. These studies also observed that vancomycin diffuses poorly in the case of ischemia. Moreover, no studies have been performed correlating local concentrations with in vitro minimum biofilm inhibitory concentrations.

Even though no specific PK studies about PK/PD and TDM of vancomycin in BJI have been performed, some studies assessing the TDM of vancomycin in patients with different types of infections have included a reduced number of patients with BJI [28]. There is also available information about the PK of intra-articular (IA)-administered vancomycin, but it is scarce and not fully understood [29]. The half-life (t_1/2_) of IA vancomycin is around 3 h (which is considerably shorter than intravenous (IV)-administered vancomycin [30]), and the clearance from synovial fluid appears to be influenced by the volume of synovial cavity, renal function and patient age [31]. Jia-Wei He et al. determined the vancomycin concentrations in serum and synovial fluid in patients with BJI treated with IV vancomycin ± IA. The two patients only treated with IV vancomycin at a dose of 1 g/12 h achieved plasma through concentrations >10 mg/mL, but no vancomycin was detected in the synovial fluid. On the contrary, patients treated with IA vancomycin reached high drug concentrations in synovial fluid but not therapeutic levels in serum. Patients treated with combined IA and IV vancomycin regimens achieved high antibiotic concentrations in synovial fluid and therapeutic vancomycin levels in serum. These results suggest that it may be difficult to achieve a high local antibiotic concentration in the site of infection with conventional IV vancomycin therapy.

After the TDM results, a dose increase was recommended in more than 30% of patients, and in around 30%, a dose reduction to avoid toxicity due to overexposure was recommended. All the recommendations were accepted by the responsible physicians.

This study has some limitations, such as the retrospective design and the lack of vancomycin levels in the synovial fluid and/or bone tissue. In addition, a potential association between PK/PD achievement and clinical outcomes could not be assessed because vancomycin treatment represents a short part of the long BJI treatment course.

## 5. Conclusions

In our study, less than 50% of patients achieved an optimal PK/PD target of vancomycin in plasma on day 2 of treatment and needed a dose adjustment. Risk factors such as age, baseline renal function, body weight and a high vancomycin dose were identified as predictors for vancomycin plasma exposure, and they should be taken into consideration in the selection of the initial dose. The individualization of the initial dose together with early TDM may allow physicians to rapidly identify patients without an optimal exposure. Future studies are needed to assess the impact of these strategies on clinical outcomes.

## Figures and Tables

**Figure 1 tropicalmed-08-00113-f001:**
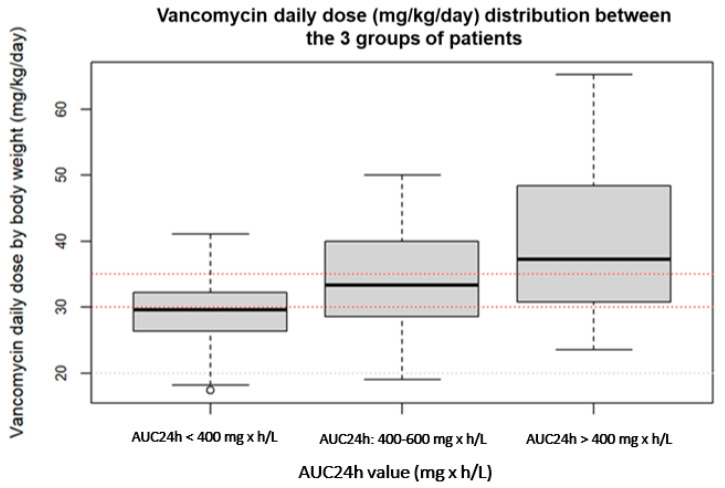
Distribution of the vancomycin daily dose values between patients with a suboptimal, optimal and supratherapeutic PKPD target.

**Table 1 tropicalmed-08-00113-t001:** Clinical and demographic data comparing patients with a suboptimal, optimal and supratherapeutic PK/PD targets.

	All Patients *n* = 94	AUC_24h_ <400 mg·h/L *n* = 22	AUC_24h_ 400–600 mg·h/L *n* = 42	AUC_24h_ >600 mg·h/L *n* = 30	*p*-Value
Gender					0.086
Male, *n* (%)	47 (50.0%)	9 (40.9%)	18 (42.9%)	20 (66.7%)	
Female, *n* (%)	47 (50.0%)	13 (59.1%)	24 (57.1%)	10 (33.3%)	
Age, years	68.5 (56.2–79.8)	61.0 (49.8–65.0)	67.5 (53.8–73.8)	82.5 (68.5–86.0)	<0.001
Body weight, kg	72.0 (61.2–87.0)	75.5 (65.0–92.5)	75.5 (65.0–90.0)	64.5 (57.2–74.2)	0.005
BMI, kg/m^2^	26.6 (23.6–31.1)	26.9 (24.7–34.2)	26.9 (24.0–31.2)	25.0 (23.3–30.1)	0.277
Obesity, *n* (%)	34 (36.2)	8 (36.4)	16 (38.1)	10 (33.3)	0.917
Charlson Index	1 (0–1)	0 (0–1)	0 (0–1)	1 (0–2)	0.034
Type of BJI, *n* (%)					0.002
FRI	23 (24.5%)	8 (36.4%)	6 (14.3%)	9 (30.0%)	
APJI	26 (27.7%)	6 (27.3%)	9 (21.4%)	11 (36.7%)	
CPJI	22 (23.4%)	3 (13.6%)	11 (26.2%)	8 (26.7%)	
OM	16 (17.0%)	2 (9.09%)	14 (33.3%)	0 (0.00%)	
Others	7 (7.45%)	3 (13.6%)	2 (4.76%)	2 (6.67%)	
Empirical treatment, *n* (%)	49 (52.1%)	14 (63.6%)	21 (50.0%)	14 (46.7%)	0.449
Directed treatment, *n* (%)	45 (47.9%)	8 (36.4%)	21 (50.0%)	16 (53.3%)	
Staphylococci, *n* (%)	38 (40.4)	7 (31.8)	19 (45.2)	12 (40.0)	0.582
MRSA	4 (4.2%)	1 (1.1%)	2 (2.1%)	1 (1.1%)	0.953
MSSA	13 (13.8%)	2 (2.1%)	6 (6.4%)	5 (5.3%)	
Others	21 (22.3%)	4 (4.3%)	11 (11.7%)	6 (6.4%)	
Enterococci, *n* (%)	7 (7.5)	0 (0)	2 (4.8)	5 (16.7)	0.055
Polymicrobial infection	20 (35.1)	1 (11.1)	10 (37.0)	9 (42.9)	0.238
CKD, *n* (%)	7 (7.45%)	0 (0.00%)	0 (0.00%)	7 (23.3%)	<0.001
Baseline SCr, μmol/L	67.2 (53.0–77.8)	64.5 (51.3–74.3)	65.4 (50.4–74.3)	70.7 (62.8–96.4)	0.030
Baseline GFR (CKD-EPI), mL/min	88.4 (32.1–147.8)	99.9 (59.3–147.8)	99.4 (44.6-139.7)	71.5 (32.1–110.1)	<0.001
Initial vancomycin dose					0.051
1 g/8 h	47 (50.0%)	6 (27.3%)	24 (57.1%)	17 (56.7%)	
1 g/12 h	47 (50.0%)	16 (72.7%)	18 (42.9%)	13 (43.3%)	
Initial vancomycin daily dose (mg/kg/day)	33.3 (27.9–40.0)	29.6 (26.4–32.1)	33.3 (28.6–39.9)	37.3 (31.4–48.0	0.001
Days of vancomycin therapy	9 (7.0–13.0)	9 (6–12)	10 (8–13.8)	8. (5.3–13.8)	0.051
SCr at the end of treatment, μmol/L	61.9 (49.5–82.8)	68.9 (54.8–71.6)	56.6 (43.3–78.7)	67.2 (43.3–78.7)	0.221
Nephrotoxicity, *n* (%)	15 (22.7%)	2 (20.0%)	4 (14.3%)	9 (32.1%)	0.275

BJI: bone and joint infection; BMI: body mass index; FRI: fracture-related infection; APJI: acute prosthetic joint infection; CPJI: chronic prosthetic joint infection; OM: osteomyelitis; CKD: chronic kidney disease; SCr: serum creatinine. Quantitative data expressed as median (interquartile range (IQR)), and categorical data as absolute frequencies (percentages). The comparison of continuous variables was performed using the one-way ANOVA test for those with a normal distribution and the Kruskal–Wallis when normality could not be assumed. For dichotomous variables, the chi-square and the Fisher exact test were applied.

**Table 2 tropicalmed-08-00113-t002:** Multivariate analysis of the risk factors for having a suboptimal PKPD target.

Variable	Odds Ratio (CI95%)	*p*-Value
Female gender	0.758 (0.231–2.487)	0.647
Age (per 10 years)	0.670 (0.446–1.005)	0.053
Vancomycin total daily dose <35 mg/kg/day	0.145 (0.033–0.638)	0.011

**Table 3 tropicalmed-08-00113-t003:** Multivariate analysis of the risk factors for having a supratherapeutic PKPD target.

Variable	Odds Ratio (CI95%)	*p*-Value
Female gender	0.213 (0.054–0.850)	0.029
Age (per 10 years)	1.580 (0.980–2.546)	0.060
Body weight (per 10 kg)	0.638 (0.386–1.054)	0.080
Baseline serum creatinine (per 0.1 mg/dL)	1.726 (1.248–2.386)	0.001
Vancomycin total daily dose >35 mg/kg/day	1.463 (0.331–6.456)	0.616

## Data Availability

The data generated are available from the corresponding author upon reasonable request.

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
