# Peer review of "Therapeutic Drug Monitoring of Vancomycin Concentrations for the Management of Bone and Joint Infections: An Urgent Need"

_tropicalmed, 2023, doi:10.3390/tropicalmed8020113_

Round 1

Reviewer 1 Report

The authors report data on vancomycin early TDM in patients treated for BJI. Some results are interesting, but several comments should be addressed to make the paper stronger.

Major comments

1/ Methods, section 2.1. Please indicate clearly how nephrotoxicity was defined and how renal function was monitored.

2/ Methods, section 2.2. The Bayesian approach used for AUC computation and dose adjustment should be explained: model used, software, residual error model, etc.

3/ Methods, section 2.3. How was vancomycin liability investigated in toxic events?

4/ Statistical analysis. I see no rationale for forcing age and sex as covariate influencing AUC target attainment. Please justify.

5/ The authors provide no results supporting the accuracy of their AUC estimation, which is critical in this study. Goodness-of-fit results should be presented.

6/ About TDM and model-based dose adjustment, please provide more results. How many dose adjustments were performed in each patient? Was the AUC target better achieved after the dose adjustment?

7/ It is surprising that the authors did not examine the relationship between exposure to vancomycin and efficacy/toxicity. This is expected as all data were available to do so, even though vancomycin is only part of the therapy.

8/ Overall, the study results and conclusions are strongly related with the initial dosing approach used in the authors’s center. The initial dose was quite fixed (1g q8h or q12h), which is surprising considering that body weight and renal function are well-known factors influencing vancomycin PK. So, finding these variables being predictive of AUC target attainment after the initial dose was expected. Most guidelines consider weight-based loading dose and renal function-based maintenance dose before TDM. The authors should mention this fixed initial dose in the abstract to better inform readers. They should also acknowledge the lack of generalizability of their results because of this dosing approach.

Minor comments

1/ Introduction: “desired PK/PD target of vancomycin is an area under the curve over 24 hours to minimum inhibitory concentration ratio (AUC24h/MIC) of 400–600 mg·h/L”.

This is not correct. The efficacy target is AUC/MIC >400 and the safety target is AUC < 600, irrespective of MIC. This leads to an AUC target of 400-600 for empirical therapy, assuming a MIC of 1 mg/L for empirical therapy. This should also be rephrased in section 2.1.

Also, the authors should provide references supporting this statement (e.g. Rybak et al.#13, Neely et al. https://doi.org/10.1128/aac.02042-17; Zasowski et al. https://doi.org/10.1128/aac.01684-17 ).

2/ Although this is reported in results, the authors should clarify the availability of MIC when vancomycin was started in the methods. Please also indicate the MIC determination method.

3/ Table 1

-          There is an error in the unit of Scr. This looks like mg/dL, not mg/L. Anyway, it would be better to use SI unit, i.e. µmol/L.

-          Which method was used to estimate GFR in ml/min?

4/ Figure 1. The legend is not correct, please revise. The boxplots show the distributions of vancomycin doses, not AUCs.

5/ Discussion. It is not true that there are no PK/PD data from patients with BJI. There have been no dedicated studies, but patients with BJI were included in various PK/PD studies. See the meta-analysis by Men et al, table 1 (https://journals.plos.org/plosone/article?id=10.1371/journal.pone.0146224 )

6/ Discussion “Surprisingly, a high initial vancomy-cin dose could not be included in the model”. This result may be explained by the arbitrary cut-off (35 mg/kg) in the analysis and the maximal initial dose capped at 3g/day. Please comment.

7/ Discussion. Please defined the “IA” abbreviation.

Author Response

Dear Reviewer,

We would like to thank you for your comments. We hope we have already answer all your questions.

Reviewer 2 Report

In this study, authors retrospectively investigated the predictive factors for achieving an optimal PK/PD target of vancomycin for bone and joint infections. As a result, more than 50% of patients did not achieve an optimal PK/PD and they needed a dosage adjustment. Although this result is interesting, the association between clinical efficacy and PK/PD has not been investigated, and the results lack clinical significance. There have been many reports that have examined factors influencing the achievement of AUC/MIC targets in conditions other than BJI, and the novelty of this paper in unclear.

Minor comments

1. Authors should describe in detail how the PK/PD parameters are calculated.

2. In microbiological detection, it is a problem that all the detected Staphylococci sp. are being examined together? MRSA and MSSA should be separated in analyzing the results.

3. It has been reported that the incidence of nephrotoxicity is associated with the trough value of vancomycin. The significance of investigating the relationship between AUC/MIC and nephrotoxicity is unclear.

Author Response

Dear Reviewer,

We appreciate your comments. We hope we have already answers all your questions.

Reviewer 3 Report

The manuscript presents the results of a retrospective study of vancomycin TDM in patients with bone and joint infections. The study provides some new knowledge, however, the manuscript requires corrections:

The abstract may be improved, especially the methods and results should be shown separately.

It is not shown how AUC24h and other PK parameters were calculated (Table 2).

More details about the program used for dose optimization should be provided.

The figure 3 caption does not reflect the presented values. It shows rather vancomycin dose ranges in all three groups of patients.

Table 3: the dose should be  > or <35 mg/kg/day?

It has not been explained whether dose adjustment led to appropriate AUC/MIC values. The Authors should show the values of AUC24h/MIC estimated after the dosage adjustment.

The statistical tests used should be added to each Table description in the Results section.

When three groups were compared it is not clear how to interpret the p values shown in the Tables.

In the Materials and methods section it is said that “A value of MIC of 1 mg/L was assumed”, whereas in the Results section that: “The median MIC of the gram-positive cocci was 1 mg/L and in 14/45 (31.1%) patients the MIC was ≥ 1 mg/L”.

Population analysis instead of the simple statistical analysis used to find factors influencing PK parameters would be more appropriate.

Author Response

Daer Reviewer,

We would like to thank you for your  comments. We hope we have already solved your questions.

Round 2

Reviewer 2 Report

none

Reviewer 3 Report

The Authors adequately addressed my concerns. I have no further comments.